# Update on Therapeutic Drug Monitoring of Beta-Lactam Antibiotics in Critically Ill Patients—A Narrative Review

**DOI:** 10.3390/antibiotics12030568

**Published:** 2023-03-13

**Authors:** Jan Stašek, Filip Keller, Veronika Kočí, Jozef Klučka, Eva Klabusayová, Ondřej Wiewiorka, Zuzana Strašilová, Miroslava Beňovská, Markéta Škardová, Jan Maláska

**Affiliations:** 1Department of Internal Medicine and Cardiology, Faculty of Medicine, University Hospital Brno, Masaryk University, 625 00 Brno, Czech Republic; 2Department of Simulation Medicine, Faculty of Medicine, Masaryk University, 625 00 Brno, Czech Republic; 3Department of Anaesthesiology and Intensive Care Medicine, Faculty of Medicine, University Hospital Brno, Masaryk University, 625 00 Brno, Czech Republic; 4Department of Paediatric Anaesthesiology and Intensive Care Medicine, Faculty of Medicine, University Hospital Brno, Masaryk University, 662 63 Brno, Czech Republic; 5Department of Laboratory Medicine, Division of Clinical Biochemistry, University Hospital Brno, 625 00 Brno, Czech Republic; 6Department of Laboratory Methods, Faculty of Medicine, Masaryk University, 625 00 Brno, Czech Republic; 7Department of Pharmacology, Faculty of Medicine, Masaryk University, 625 00 Brno, Czech Republic; 8Department of Clinical Pharmacy, Hospital Pharmacy, University Hospital Brno, 625 00 Brno, Czech Republic; 92nd Department of Anaesthesiology University Hospital Brno, 620 00 Brno, Czech Republic

**Keywords:** beta-lactam antibiotics, pharmacokinetics–pharmacodynamics, critically ill, bacterial susceptibility, therapeutic drug monitoring, high-performance liquid chromatography

## Abstract

Beta-lactam antibiotics remain one of the most preferred groups of antibiotics in critical care due to their excellent safety profiles and their activity against a wide spectrum of pathogens. The cornerstone of appropriate therapy with beta-lactams is to achieve an adequate plasmatic concentration of a given antibiotic, which is derived primarily from the minimum inhibitory concentration (MIC) of the specific pathogen. In a critically ill patient, the plasmatic levels of drugs could be affected by many significant changes in the patient’s physiology, such as hypoalbuminemia, endothelial dysfunction with the leakage of intravascular fluid into interstitial space and acute kidney injury. Predicting antibiotic concentration from models based on non-critically ill populations may be misleading. Therapeutic drug monitoring (TDM) has been shown to be effective in achieving adequate concentrations of many drugs, including beta-lactam antibiotics. Reliable methods, such as high-performance liquid chromatography, provide the accurate testing of a wide range of beta-lactam antibiotics. Long turnaround times remain the main drawback limiting their widespread use, although progress has been made recently in the implementation of different novel methods of antibiotic testing. However, whether the TDM approach can effectively improve clinically relevant patient outcomes must be proved in future clinical trials.

## 1. Introduction

Antibiotics are one of the most prescribed types of medication in intensive care. In a large European observational study, 64% of patients were exposed to antibiotics during their ICU stay [1]. There are several factors contributing to this fact. Patients with severe infection usually need intensive care; moreover, sepsis alone is one of the leading causes of admission to intensive care [2]. In addition to sepsis, critically ill patients hospitalized for any other reason (major trauma, post-cardiac arrest, major surgery, etc.) are prone to developing serious infections. 

Key components of adequate antibiotic treatment are the selection of the appropriate antibiotic (or combination of antibiotics), timely administration and adequate dosing. Both delays in antibiotic therapy and insufficient dosing could result in increased mortality [3,4]. During past few decades, great progress has been achieved regarding timely antibiotic administration in septic patients worldwide, mainly due to the Surviving Sepsis Campaign guidelines being incorporated into general medical knowledge [3,5]. Despite increasing levels of bacterial resistance worldwide, beta-lactam antibiotics have remained a preferred group of antibiotics due to their availability and excellent safety profile [6,7]. They demonstrate bactericidal activity, acceptable tissue penetration and relatively low toxicity compared to other groups of antibiotics [8]. 

Several aspects of administering beta-lactam antibiotics should be considered, such as the site of infection, the immune status of the patient, the severity of infection, the level of renal impairment and the use of renal replacement therapy (RRT) or extracorporeal membrane oxygenation (ECMO). Antibiotic dosing adjustments in such patients are particularly challenging. The underdosing of beta-lactam antibiotics is reported in many critically ill patients [9,10,11]. This presents a dangerous threat, as clinical effectiveness may be strongly diminished and the development of bacterial resistance promoted. On the other hand, the risks of drug toxicity increase after exceeding defined plasmatic levels [12,13]. 

Our review describes various aspects of beta-lactam use in critical care, pharmacokinetics/pharmacodynamics (PK/PD), the characteristics of beta-lactams and the main factors of PK/PD variability in critically ill patients. Different modes of application, such as prolonged infusions and continuous infusions, are discussed. Current evidence supporting antibiotic guidance based on therapeutic drug monitoring (TDM) in critically ill patients is analyzed and, finally, recommendations on beta-lactam dosing utilizing target drug monitoring are given.

## 2. Beta-Lactam PK/PD General Characteristics

Achieving the adequate concentration of any antibiotic at the site of infection and preventing bacterial resistance is crucial for good clinical practice. The knowledge of the pharmacodynamics and pharmacokinetics of any antibiotic is essential for formulating an optimal dosing regimen. Different groups of antibiotics demonstrate various PK/PD properties. The clinical effectiveness of beta-lactams is based on the time that their unbound fraction spent above the minimum inhibitory concentrations (MICs) of the susceptible microorganisms [14,15]. This phenomenon is called time-dependent killing [16]. Beta-lactam antibiotics are small hydrophilic molecules with a low volume of distribution (V_d_) characterized by tissue distribution limited to the extracellular space, i.e., plasma and interstitial compartment. This results in limited penetration across biological barriers. Binding to plasma proteins is significant in cephalosporins except for cefotaxime, ceftaroline, ceftolozane and cefepime. Ertapenem, flucloxacillin and oxacillin also display high binding to plasma proteins (>30–50%), while the free fraction of the remaining beta-lactams exceeds 80%. Apart from ceftazidime, ceftriaxone, ceftaroline and ceftolozane, the elimination half-life of beta-lactams is less than 2 h. The PK/PD characteristics of beta-lactam antibiotics are summarized in Table 1. Since most of these antibiotics are excreted primarily via glomerular filtration in kidneys, renal functions are critical factors affecting antimicrobial concentration [17]. 

## 3. Factors Affecting Beta-Lactams PK/PD Targets in Critically Ill Patients

Specific dosing regimens are often derived from the pharmacokinetic studies of a population of healthy volunteers, although the physiology of critically and non-critically ill patients differ. Thus, the standard dosing of antimicrobials may be inadequate, not reaching the PK/PD target, particularly in earlier phases of critical illness [18]. Critically ill patients display many alterations in their physiology, affecting the PK/PD of many molecules, including beta-lactams. Factors contributing to changes in pharmacokinetics during critical illness are as follows: increased V_d_, decreased plasma binding capacity due to hypoalbuminemia and altered excretion by the kidneys and, to a lesser extent, by the liver [19,20]. The increased volume of distribution and altered drug clearance depends on renal function, representing the two critical factors affecting antibiotic concentrations in plasma and subsequently at the site of infection [17]. An important physiological alteration affecting the excretion of beta-lactams, found in up to two-thirds of critically ill patients, is augmented renal clearance (ARC) [21,22]. ARC, defined as creatinine clearance (CL_cr_) greater than 130 mL/min, has been described in a considerable proportion of early septic patients [22]. The precise mechanism has not been completely elucidated but increased cardiac output with changes occurring in renal microvasculature has been described [23].

The systemic inflammation and following management of sepsis have several important pathophysiological consequences affecting drug metabolism. First, endothelial dysfunction results in large quantities of intravascular fluid moving into interstitial tissue, increasing the V_d_ of any hydrophilic drug including beta-lactams [24]. Second, aggressive fluid therapy by the infusion of large volumes of crystalloids can promote this consequence further by reducing the arterial load [25]. Third, generalized vasodilatation leads to relative intravascular hypovolemia and using only large-volume fluid treatment can induce a vicious cycle of progressive shock, tissue oedema and organ dysfunction [26]. Fourth, mechanical ventilation, post-surgical drainage and hypoalbuminemia contribute to the further expansion of V_d_ [27,28]. These factors play a key role in decreased antibiotic concentration in plasma and, consequently, at the site of infection. In a large multinational trial investigating the pharmacokinetics of eight beta-lactam antibiotics during critical illness, the considerable variability of achieved serum concentrations was found. More importantly, low plasma concentrations resulted in suboptimal clinical outcomes, defined as a need for changing the antibiotic or adding another antibiotic within the treatment period or within 48 h of completion [9].

In septic shock, microvascular failure may decrease antibiotic concentrations at the infection site [29]. Measuring antibiotic concentrations directly in the infected tissue is often impossible or may present a significant risk to the patients and is thus not performed routinely [30]. Hence, the only surrogate for estimating tissue concentrations is the plasmatic concentration [31]. When applying PK/PD concepts, only the unbound fraction of the drug can pass through biological membranes, diffuse into tissues and thus exhibit antimicrobial effects [32]. Liver dysfunction plays an important role in the elimination of beta-lactams excreted primarily hepatobiliary, e.g., cefoperazone [20]. Furthermore, the decreased hepatic synthesis of albumin may further contribute to changes in pharmacokinetics in critically ill patients.

These changes from normal physiology are often superimposed on a chronic condition, further complicating the predictability of drug metabolism and clearance [33]. Simultaneously, the level to which these factors play role in an individual patient usually varies over the time course of a critical illness [34]. Thus, to achieve the optimal effectiveness of any antibiotic used in a critically ill patient, it is of paramount importance to consider all these factors. 

## 4. Consideration in Patients with Hypoalbuminemia

Hypoalbuminemia is present in 40–50% of critically ill patients and may be associated with an increased free fraction of highly protein-bound beta-lactams [27,31]. Apart from ceftazidime, cefotaxime and cefepime, most cephalosporins exhibit high binding to plasma proteins [35]. A high level of protein-binding was first described in penicillin and carbapenems and later in ertapenem and derivates of oxacillin [15,36]. In critically ill patients, albumin levels correlate with the severity of illness [37], leading consistently to hypoalbuminemia. The resulting plasma binding capacity decreases and the PK of these drugs is altered. Increased free fraction enables better antibiotic penetration into the tissues but, simultaneously, improved renal clearance leads to their shorter plasmatic half-life. Moreover, the co-administration of other drugs with high protein-binding capacities will further promote a free fraction of beta-lactams, making the situation even more complex [15]. When using beta-lactams highly bound to plasma proteins, such as ceftriaxone or ertapenem, the loading dose in patients with severe hypoalbuminemia should be increased [38]. For TDM purposes, the direct measurement of the unbound fraction of the antibiotic using the ultrafiltration method followed by high performance liquid chromatography (HPLC) has been developed [39,40]. Briefly, HPLC is an analytical method used to separate, identify and quantify each component in a mixture. It utilizes different flow rates of the components through a column filled with the solid adsorbent, resulting in the separation of the components as they flow out of the column [41]. However, unfortunately, using this combination of ultrafiltration with HPLC in clinical practice is limited. 

## 5. Considerations for Patients with Renal Dysfunction

The glomerular filtration rate (GFR), particularly in patients with sepsis/septic shock, can be significantly altered in both directions. A significant proportion of critically ill patients show signs of ARC, but on the other hand, acute kidney injury (AKI) with decreased GFR is one of the most common organ dysfunctions in an ICU setting. This makes antibiotic dosing in critically ill patients difficult [42]. Antibiotic dosing in patients with acute kidney injury and chronic kidney disease, especially while on renal replacement therapy, presents a particular challenge [43]. 

Acute kidney injury (AKI) is one of the most important organ dysfunctions. In critically ill patients, the incidence of AKI ranges from 30 to 60% [44]. Its clinical manifestation covers only mild elevations of serum creatinine to severe metabolic consequences, including hyperkalemia and metabolic acidosis. The pathophysiology of AKI, especially in patients with sepsis/septic shock, is complex. Its evolution depends on various factors, such as the adequate source control of the infection, the immune status of the patient, the administration of nephrotoxic drugs and the intensity of fluid therapy [45]. Estimates of glomerular filtration based on patients with chronic renal disease, such as the Cockroft–Gault and the modified diet in renal disease (MDRD) equations, do not represent the actual level of kidney dysfunction in critically ill patients [46]. Serum creatinine levels are not able to reflect dynamic changes in renal function. So, dose adjustments determined by single creatinine value are inappropriate in this setting. Although cystatin C as a biomarker was shown to have better performance in determining GFR in critically ill patients [47], creatinine clearance (CL_cr_) still represents clinically the most adequate estimation of the true GFR value. Even when based on shorter periods of urine sampling, it reflects GFR variation well [48].

About 5% [49] of critically ill patients with AKI need renal replacement therapy. There are three principal options regarding the length and intensity of the procedure. Regional differences regarding clinicians’ preferences exist [50], so all three methods will be briefly introduced. The first is continuous renal replacement therapy (CRRT), comprising continuous veno-venous hemofiltration (CVVH), continuous veno-venous hemodialysis (CVVHD) and continuous veno-venous hemodiafiltration (CVVHDF). The other modalities are classical intermittent hemodialysis (IHD) and sustained low-efficiency dialysis (SLED), sometimes referred to as prolonged intermittent renal replacement therapy (PIRRT). 

The main mechanisms of drug clearance by RRT are convection, diffusion and adsorption, all of which play a role in beta-lactam clearance due to their small molecule and low protein binding. The dialysate flow rate, blood flow rate, material and surface area of the hemofilter and the ultrafiltration rate and length of the procedure play role in beta-lactam removal [51]. Apart from factors related to the procedure itself, several patient characteristics have a direct influence on the PK of beta-lactam antibiotics during RRT. As already mentioned, most beta-lactams have only a small fraction bound to plasma proteins, mainly albumin. Thus, hypoalbuminemia may promote beta-lactam clearance by increasing its free fraction [52]. Residual renal function in patients with preserved urine output should also be considered, as it can contribute significantly to antibiotic clearance [53,54]. Creatinine clearance measurement based on urine collected during periods between dialysis procedures is a simple method of residual kidney function assessment.

On the other hand, [22,55] ARC, i.e., augmented glomerular clearance, determined using CL_cr_, can lead to suboptimal serum levels of beta-lactams, [56] although CL_cr_ may overestimate the true value of GFR [55,56].

Likewise, in AKI, serum creatinine is an unreliable marker of these changes [57], so the clinical suspicion of this syndrome should prompt the investigation of the creatinine clearance. 

As a result, the PK of beta-lactams in patients with renal disease, and particularly in those receiving any form of RRT, is highly unpredictable. As shown recently, the dosing regimens of beta-lactams in critically ill patients treated with CRRT are highly variable, reflecting the need for fitting the therapy to the actual needs of the patient [58]. To achieve an optimal PK profile, TDM is recommended in such patients [14,15]. 

## 6. Considerations for Patients on Extracorporeal Life Support

Extracorporeal membrane oxygenation (ECMO) is important organ support provided to critically ill patients with severe respiratory or circulatory failure. Its use has expanded in the past several years especially, due to the COVID-19 pandemic [59]. The significant influence of ECMO itself on the PK/PD characteristics of most beta-lactams has been described [60]. In a monocentric German study, the plasmatic concentrations of beta-lactams were found to be profoundly decreased during ECMO procedures [61]. While awaiting further data, the TDM approach in patients on ECMO seems necessary. 

## 7. Biochemical Assays for TDM of Beta-Lactam Antibiotics

Routine antibiotic analysis in clinical laboratories is usually limited to aminoglycosides and vancomycin to prevent their nephro- and oto- toxic effects [62]. These antibiotics are analyzed by various immunoassays, such as the kinetic interaction of microparticles in solution (KIMS), cloned enzyme donor immunoassay (CEDIA), and particle-enhanced turbidimetric inhibitor immunoassay (PETINIA), to name a few [63]. These methods are performed on clinical chemistry automated analyzers—standard equipment in all clinical laboratories developed and maintained by in vitro diagnostics (IVD) companies. The main advantage of immunoassays is their fast implementation in the laboratory. Since reagents come in ready-to-use packs, laboratory staff require only short training and automated analyzers enable the high throughput of samples, as all tests are performed in parallel. From a clinician’s point of view, the main advantage is the fast turnaround time (TAT). However, due to high initial costs, only a limited portfolio of antibiotics is currently available for TDM immunoassay analysis [64]. Due to the variability of companies with a portfolio of analyzers based on a similar analysis principle, these tests can be performed in most current clinical laboratories. On the other hand, immunoassays provide a greater chance of possible interferences and cross-reactions, resulting in false results.

An increased range of antibiotics for TDM may be attained by utilizing chromatographic methods, predominantly high-performance liquid chromatography (HPLC) coupled with ultraviolet (UV) [65] or mass spectrometry (MS) [66] detectors. Several companies have developed methods of ready-to-use kits for the quantitative analysis of several antibiotics, including beta-lactams, in plasma. However, since instrumentation in different laboratories vary, the method transfer is more complicated and time-consuming than with the previously described immunoassay methods [67]. The low throughput of the analyzers is another disadvantage of the chromatographic methods. Patient samples, as well as internal controls and calibrators, must first undergo a timely and complicated extraction process. The analysis is then performed in tandem. This leads to long TATs, which diminish the effectiveness of the TDM process. The advantages are the possibility of simultaneous analysis of several analytes [68] and robust results with high specificity and sensitivity. Another advantage of chromatographic methods is the ability to develop methods on-site (in-house), so the laboratories can provide a larger portfolio of analytes. However, in-house method development and method validation is very time consuming and requires skilled analytical personnel [64,69]. 

Quite recently, the analysis of ceftazidime and piperacillin via immunoassay on a tabletop analyzer was introduced, ensuring short TAT and allowing a point-of-care (POC) setting. Another feasible approach may be the automation of the HPLC methods coupled with a mass analyzer [70]. 

## 8. Microbiological Susceptibility Testing

The determination of MIC has gained a reputation as the golden standard of antibiotic stewardship over the past decades [71]. When trying to achieve a defined plasmatic level of an antibiotic based on minimal inhibitory concentration testing, the clinician must understand the potential drawbacks of this approach. MIC is provided by the in vitro testing of the inhibition of bacterial growth in standardized inoculum on standardized media using a defined assay. It utilizes two-fold dilution above and under the antibiotic concentration of 1 mg/L, so the resulting concentration is typically expressed as one value from the interval (0.002, 0.004, 0.032… 256, 512) mg/L [72]. The first shortcoming arises from the time needed for culture and testing. As it reaches at least 48 h in most settings, it leaves questions regarding the adequacy of the treatment for quite a significant time [73]. The second drawback of MIC testing stems from the variability of the results themselves. Bacterial strains without any acquired resistance, so-called wild type (WT) bacteria, demonstrate distribution around three to five two-fold concentrations. The highest WT concentration within the limits of natural distribution is defined as the epidemiological cut-off (ECOFF). These data are regularly updated by the European Committee on Antimicrobial Susceptibility Testing (EUCAST) and are available online [74]. Another source of MIC variation is caused by assay differences. It has been shown that if MICs are determined several times in more than one laboratory, over half of the variability is due to strain-to-strain variation and inter-laboratory differences, with the remainder being attributed to the assays themselves [75]. 

The PK/PD objective of beta-lactams is expressed as the percentage of time, during which the free antimicrobial’s fraction in plasma exceeds a certain concentration (% fT > concentration). Hypothetically, the target concentration should be equal to the MIC of the treated pathogen. Considering MIC variability, the use of one single MIC value obtained by MIC determination for achieving 50–100% fT > MIC is rather inappropriate and higher PK/PD targets seem necessary. To prevent antibiotic underdosing regarding MIC variability, an individualized approach was suggested. If MIC falls into a low-level resistance area, indicating that the strain is within the WT distribution range, the upper ECOFF value should be taken as a target. If the MIC is above the upper ECOFF value but still below the clinical resistance breakpoint, the PK/PD target should be set as MIC + two-fold dilution, meaning four times the MIC value for the 100% of the dosing interval (fT ≥ 4 × MIC = 100%). If the MIC value that is clearly above the clinical resistance breakpoint, switching to a different antibiotic is a clear option [76]. 

Assays using a modified MIC approach were developed to assess serum antibiotic levels. The main advantage comes with lower costs and technological requirements. From their introduction in voriconazole [77], microbiological methods have proven effective in determining cefotaxime, meropenem and piperacillin, resulting in strong correlations with values obtained by HPLC [78]. 

## 9. Beta-Lactam Toxicity

Clinically relevant beta-lactam toxicity comprises effects on the central nervous system, hepatotoxicity, myelosuppression, nephrotoxicity and *Clostridoides difficile* infection (CDI) [79]. Over the past few decades, the neurotoxic effects of beta-lactams have become more familiar among clinicians. The reported clinical manifestation ranges from electroencephalographic changes, the altered quantitative level of consciousness, confusion, hallucinations, movement disorders (asterixis, dyskinesis), myoclonus and, most importantly, seizures or even status epilepticus [13,15]. The overall incidence of beta-lactam-related neurotoxicity remains debatable. In patients treated with cefepime, piperacillin/tazobactam or meropenem, up to 15% experienced signs of neurotoxicity [13]. Nevertheless, in a recent retrospective cohort study, the overall incidence was found to be between 2.1% and 2.6% [80]. The greatest potential for inducing seizures was described in cefazolin and cefepime, followed by penicillin G and imipenem [79]. As beta-lactams cross the blood–brain barrier, a direct relationship between high plasmatic concentrations and neurotoxicity was found. Renal dysfunction with an unpredictable increase in plasmatic and tissue concentrations of beta-lactams presents a major risk factor, with a history of neurological disorders also being a predisposing factor [79,81,82]. Potential toxicity mediated by concentrations (when applied by discontinuous infusions) and steady-state (in case of continuous infusions) concentrations in plasma were identified for flucloxacillin, amoxicillin, ceftazidime, piperacillin/tazobactam, cefepime, imipenem and meropenem [12,13,15,83,84]. 

The nephrotoxic effects of beta-lactams remain underrated but still debated. The risk of nephrotoxicity is even higher when combined with certain known nephrotoxic drugs, e.g., vancomycin, especially in patients with premorbid kidney disease or older age [79,85]. Although the reported incidence is highly variable [79] and direct causality can be found only rarely, the deterioration of kidney functions puts critically ill patients at a greater risk of death [49]. Despite epidemiological data showing an association of AKI with beta-lactam administration, direct causality can be found only rarely. A possible increase in AKI related to the combination of antibiotics, for example, the most frequently used piperacillin and vancomycin, is not based on evidence of causality [86]. Surprisingly, data suggesting the protective role of a combination of piperacillin and vancomycin exist [87,88]. Additionally, these conflicting data regarding nephrotoxicity are based on serum creatinine increase [89]. The proximal tubular secretion of creatinine could be reduced by several antibiotics, including piperacillin or vancomycin. They bind with higher affinity to renal transporters mediating creatinine secretion and, consequently, serum creatinine levels increase. Thus, the association with AKI defined by creatinine levels should be called pseudotoxicity, rather than defined as a real toxic effect [86]. As already mentioned, using a single creatinine level in estimating GFR in a critically ill patient is not appropriate, and other approaches, such as measuring cystatin levels or four-hour creatinine clearance, should be prioritized [90,91]. 

Acute interstitial nephritis is the usual underlying mechanism with non-IgE mediated hypersensitivity reaction and T-lymphocyte involvement [79,92]. When clinical suspicion is supported by skin rash and microscopic hematuria with proteinuria, corticosteroids represent a therapeutic option [93]. 

Myelosuppression with severe neutropenia is a rare but potentially fatal complication of beta-lactam exposure, usually resolving after discontinuation of the treatment [94]. Cross-reactions after the institution of different beta-lactam antibiotics have also been described.

As the toxic effects of beta-lactams are directly related to their plasmatic concentration, the upper limit of plasmatic concentration 8 × MIC should not be exceeded [14,15]. 

## 10. PK/PD Targets for Beta-Lactam Antibiotics

As mentioned earlier, the PK/PD target directly connected to the bactericidal effect of beta-lactams is expressed as the percentage of time during which the free antimicrobial’s fraction in plasma spends above a certain level (% fT > concentration). As the post-antibiotic effect of beta-lactams is variable, the peak plasma concentration has no significant benefits [17,32,95] and is not standardly accounted for. The required concentration of a particular beta-lactam antibiotic is dependent upon the MIC of the causative pathogen. Based on experimental data, the PK/PD index associated with optimal beta-lactam activity was defined as fT > MIC at 50–70% for most infections [14]. However, maintaining the concentration above MIC 100% of the time was shown to be associated with better outcomes in critically ill patients [9,96]. When taking into account microbiological testing variability and inconsistent penetration into infected tissues, even higher PK/PD targets are preferable. To deal with all sources of individual variability, a concentration four times higher than the MIC for 100% of the dosing interval should be achieved to optimize clinical outcomes and, at least, to prevent the selection of resistant bacterial subpopulations [15,97]. Whether this approach helps improve clinical outcomes is not yet proven; moreover, the DALI study was not able to show the utility of the PK/PD concepts in the clinical setting of this trial [9]. Considering the threshold for toxicity, the target concentration of beta-lactam antibiotics should be between four- and eight-times above MIC for 100% of the time (fT ≥ 4–8 × MIC = 100%) [15].

## 11. Modes of Applications of Beta-Lactam Antibiotics

For beta-lactams, as typical time-dependent killing antibiotics, optimal PK/PD targets of beta-lactams are achieved by keeping the plasmatic concentration within certain concentration limits without major fluctuations. Based on population pharmacokinetic studies, extended-length (usually ≥ 3 h) or continuous infusions following a loading dose provide better attainment of PK targets than standard infusions [98,99,100]. The clinical benefit was proven in patients with severe sepsis [101], and although this finding may not be consistent [102], results of meta-analyses suggest better outcomes in septic patients treated with this strategy [103,104,105]. These outcomes were most prominent in critically ill or immunocompromised patients with infections caused by non-fermenting Gram-negative bacteria, especially *Pseudomonas aeruginosa* [106,107]. Prolonged infusion resulted in improved outcomes in patients with lower respiratory tract infections [103,108]. The administration of beta-lactams in prolonged or continuous infusions has also been recommended in the latest Surviving Sepsis Campaign guidelines [5].

The chemical stability of beta-lactam infusions lasting more than several hours has been questioned. This becomes an issue with imipenem (2 h), meropenem and ertapenem (6 h). Other beta-lactams remain stable after reconstitution in a 0.9% NaCl solution for more than 8–12 h, enabling their safe use in form of a continuous infusion with several changes of a new antibiotic solution per day [109,110].

When applying a beta-lactam antibiotic in the form of a continuous or prolonged infusion, the application of a loading dose is crucial [15,106]. The optimum initial dose for each antibiotic is calculated primarily by its V_d_ and should not be modified according to the degree of organ dysfunction. The administration of a loading dose identical to the dose used in intermittent application seems to be a reasonable approach, see Table 1.

## 12. Considerations in Pediatric Patients

Antibiotics are one of the most used medications in hospitalized pediatric patients. The reported use in a pediatric setting is prevalent in between 40 to 79% of patients/cases [111,112,113,114,115] but even higher in pediatric intensive care units (PICU). The reported inadequate antibiotics administration reaches up to 60% in PICU settings (type, dose, duration) [116,117]. Antibiotic overuse in pediatric patients seems to be frequent. For example, unindicated prolonged antibiotics prophylaxis was described in up to 78% of patients [118,119,120]. These factors could lead to inappropriate clinical effects, toxicity issues or antibiotic resistance. In pediatric pharmacology, clinicians are dealing with a lack of data regarding the effective dose and safety profiles in a huge proportion of daily administered medications (off-label administration) [121]. Only a minority of routinely used medications have been tested in properly designed and powered clinical trials with pediatric patients for their pharmacokinetics/pharmacodynamics properties and almost none in critical care settings. Almost all aspects of pharmacokinetics (PK) are affected in critical illness, and the drug dose optimization process is necessary in order to reach the effective antibiotic plasmatic concentration. Maximum PK alterations have been identified in early childhood due to rapid growth [116,122,123]. When considering the proportion of renal drug elimination, greater clearance and glomerular filtration is described in children compared to adults [124,125].

Besides PK changes caused by critical illness or therapeutic processes (non-maturational), developmental and maturational changes (age, weight, organ functional status—e.g., kidney excretion) during childhood also significantly affect the PK antibiotics profile [111]. These factors explain the significant PK variability in pediatric patients [111,126,127]. The simple weight-based or age-based dose adjustment led to inadequate plasmatic antibiotic levels [111,128,129]. The antibiotics stewardship principles together with TDM have been implemented in many hospital settings to solve the potential overuse of antibiotics and to optimize the dose and duration of the treatment in inpatients. The optimal appropriate antibiotics treatment (type, dose, duration), including PK/PD adjustments, has been associated with a positive outcome of critically ill patients [9,125,130,131,132]. Antibiotic stewardship, target drug monitoring and close cooperation with clinical pharmacist could optimize this process.

Standard β-lactams dosing in PICU settings has been associated with a 95% incidence of underdosing [125], which is an objective risk factor for antibiotics resistance [133] and/or reduced efficacy of treatment (morbidity and mortality risk). TDM approach in critically ill pediatric patients could be further upgraded with model-informed precision dosing (MIPD), where the dosing process (initial dose, time of infusion, timing, maintenance) is individualized by a specific commercial-based algorithm [134]. Algorithms that can be applied in pediatric patients include Auto Kinetics, Precise PK and BestDose [134]. Beside algorithm-based antibiotic dosing, antibiotic stewardship programs and close cooperation with clinical pharmacists could optimize antibiotic management (dose, duration, type).

## 13. Therapeutic Drug Monitoring of Beta-Lactam Antibiotics

Despite large body of data suggesting the better attainment of PK/PD targets using the monitoring of the plasmatic levels of beta-lactam antibiotics [10,135,136,137], better clinical outcomes by utilizing TDM have not been seen so far. High-quality data from prospective randomized clinical trials (RCTs) are scarce, while three multicenter RCTs prospectively evaluating the effectiveness of the TDM approach in critically ill patients have been published recently. The TARGET trial assessed the TDM of piperacillin/tazobactam in septic patients and, despite the better achievement of PK/PD targets, no significant improvement in organ dysfunction was found [138]. The DOLPHIN trial evaluated MIDP (TDM coupled with dosing software) in patients treated with beta-lactams or ciprofloxacin. No benefit in terms of shortening the length of stay was observed, while target attainment in both MIDP and control groups was similar [139]. In a two-center Dutch trial, the adaptive dosing of four antibiotics including meropenem and ceftriaxone in septic patients did not lead to the better target attainment of these beta-lactams nor did it improve any clinical outcome [140]. While the effectiveness of the TDM guided approach to beta-lactam prescription has not been proven, it may be beneficial in certain subgroups of critically ill patients. The selection of eligible patient subgroups remains a main challenge for future research [141].

At present, the measurement of plasmatic beta-lactam concentrations should be considered in patients with sepsis, patients with pneumonia and in patients infected with Gram-negative pathogens and with bacteria with decreased susceptibility [9,15,142]. Further TDM is mandated in patients with presumed high variability in pharmacokinetics and those with AKI, especially on RRT [15,40]. The measurement of plasmatic concentration is also clinically justified when experiencing potential signs of toxicity [79]. When administering beta-lactams in the form of a continuous infusion, steady state concentration after 24–48 h of infusion is recommended. When applying discontinuous infusions, trough concentration after steady state institution should be evaluated. As the plasmatic half-life of beta-lactams generally does not exceed several hours, it is reasonable to apply TDM after 24–48 h of application. New measurements of plasmatic levels seem to be required whenever the clinical status changes (new organ dysfunctions, signs of possible toxicity). Long turnaround times of most analytical methods remain a limitation of this approach. Nevertheless, having the results available within 24 h seems a reasonable compromise in most clinical scenarios.

A pragmatic approach to the target drug management of beta-lactam antibiotics is outlined in Figure 1. Characteristics of the most widely used beta-lactam antibiotics are summarized in Table 1.

## 14. Conclusions

Despite significant advances in sepsis treatment and improvements in the care of critically ill patients over the last few decades, sepsis mortality remains well above 20% [146]. Antibiotics are key components of sepsis therapy and daily clinical routine in the ICU. Unfortunately, the target attainment of beta-lactam plasmatic concentrations remains unsatisfactory in 40–60% of critically ill patients [9,142]. Following the general trend of personalizing therapy to the specific patient, therapeutic drug monitoring is a highly useful tool for dose optimization. TDM helps in minimizing risks of under- or overdosing. Although the body of evidence supporting its usefulness in antibiotic prescription is growing, the widespread utilization of TDM in clinical practice is still far from reality. Although recent data do not support the TDM approach in unselected critically ill patients, antibiotic stewardship utilizing therapeutic drug monitoring in different subgroups of critically ill patients, namely in those with septic shock and acute kidney injury requiring any form of renal replacement therapy, seems still necessary. Shorter turnaround times of novel analytical methods for antibiotic concentrations will hopefully lead to the wider implementation of these techniques into clinical use.

## Figures and Tables

**Figure 1 antibiotics-12-00568-f001:**
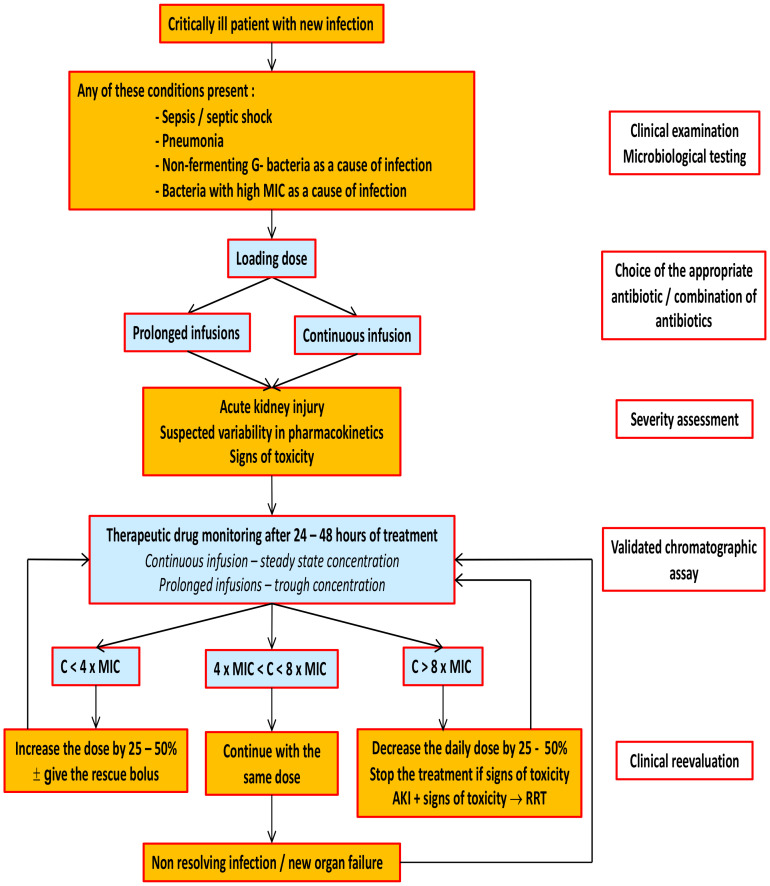
Simple algorithm utilizing therapeutic drug monitoring. “High MIC” for a given beta-lactam antibiotic denotes a MIC value above the median of the distribution of MIC values for wild-type strains of the considered bacteria. G-bacteria: Gram-negative bacteria. Based on recommendations published previously [15].

**Table 1 antibiotics-12-00568-t001:** General characteristics of beta-lactam antibiotics.

Antibiotic	MW (Da)	Free Fraction (%)	t_1/2_ (Hours)	V_d_ (L/kg)	Initial Dose for Critically Ill Adults	Toxicity Threshold
ampicillin/sulbactam	581/255	85/62	1.2/1	0.29/0.25	3 g	>8 × MIC
amoxicillin/clavulanate	365/199	82/75	1–1.4/1	0.36/0.21	1.2 g	c_min_ > 40 mg/L
oxacillin	401	6–10	0.5–0.7	0.4	2 g	>8 × MIC
flucloxacillin	454	4–5	0.5–1	2.18	2 g	c_min_ > 125 mg/L
piperacillin	518	70	1	0.24	4 g	c_min_ > 361 mg/L
piperacillin/tazobactam	518/300	70/78	1/1	0.24/0.40	4.5 g	c_min_ > 64 mg/L c_ss_ > 157 mg/L
cefazolin	454	20	2	0.19	2 g	>8 × MIC
cefoxitin	427	21–35	1	0.23	2 g	>8 × MIC
cefuroxime	424	50–67	1.5	0.19	1.5 g	>8 × MIC
ceftazidime	547	90	2.8	0.28–0.40	2 g	>8 × MIC
ceftriaxone	554	10	5–9	0.1–0.2	2 g	>8 × MIC
cefotaxime	455	50–70	1.5	0.28	2 g	>8 × MIC
ceftaroline	684	80	2.7	0.29	600 mg	>8 × MIC
ceftolozane/tazobactam	666/300	80/78	3.1/1	0.19/0,40	3 g	>8 × MIC
cefepime	481	84	1.7–2.3	0.3	2 g	c_ss_ > 35 mg/L c_min_ > 20 mg/L
meropenem	383	98	1	0.35	2 g	c_min_ > 64 mg/L
imipenem	317	80	1	0.22	1 g	>8 × MIC
doripenem	420	92	1	0.24	1 g	>8 × MIC
ertapenem	475	20–40	4	0.12	1 g	>8 × MIC

MW(Da)—molecular weight (Dalton); t_1/2_—plasmatic half-life; V_d_—volume of distribution; c_ss_—steady-state plasmatic concentration (when applying continuous infusion); c_min_—plasmatic trough concentration (in discontinuous application) [Derived from [12,13,15,84,143,144,145]].

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
