# Peer review of "Update on Therapeutic Drug Monitoring of Beta-Lactam Antibiotics in Critically Ill Patients—A Narrative Review"

_antibiotics, 2023, doi:10.3390/antibiotics12030568_

Round 1
Reviewer 1 Report
The authors reviewed PK/PD characteristics of beta-lactams and presented potential strategies that may help to improve the management of this drugs in critically ill patients with infection and sepsis. Although the authors shed light on a interesting topic, this manuscript is sometimes unclear and potentially misleading thus warranting MAJOR revision.
Please, summarise the PK characteristics of beta-lactams (e.g. hydrophylicity and protein binding);
Page 2, lines 93: the kidney is not the sole organ where antimicrobial metabolism takes place. Please, report eventual implications of liver dysfunction and extracorporeal organ support therapies on affecting PK characteristics of antimicrobials
Page 3, lines 141-144: please support this concept with scientific evidence otherwise you kindly asked to remove it.
Page 3, line 140: provide a brief description of HPLC
Page 3, lines 147-151 and Page 4, lines 160-161: this section is conceptually wrong and potentially misleading. Please, rewrite it and provide some appropriate reference to support what you wrote.
Page 4, line 175: please, add absorption otherwise this is imprecise.
Page 4, lines 183: explain what residual renal function is and how clinicians can assess it
Finally, you are kindly asked to make clear what is the role of TDM and its implications on patient outcome. Moreover, you should compare its use with standard antimicrobial dosing based on empiric rules otherwise your manuscript appears purely speculative and not supported by scientific evidence.
Page 5, lines 217-219: provide references
Page 5, "microbiological susceptibility testing": this section is very hard to read and must be improved by including a brief description of PK7PD targets, otherwise "fT" doesn't make any sense.
Page 6, "beta-lactam toxicity": please, remove any description of the pathophysiology of beta-lactam neurotoxicity. It is not the topic of your paper.
Page 7, line 320: provide references
Page 7, line 342-345: please explain the importance of the loading dose and how it is computed (volume of distribution). Moreover, say whether it should be modified according to the severity of organ dysfunction (NO!).
Page 8, lines 399-400: what do you mean?
Reviewer 2 Report
Update on therapeutic drug monitoring of beta-lactam antibiotics in critically ill patients - a narrative review
In general, it is a comprehensive and well written and organized narrative review that includes the most important references published on the topic. The diagram is interesting.
Commentaries to the authors
- Lines 27-29 may be rethought. To date, the published evidence does not clearly support the minimization of the risks resulting from antibiotic under and overdosing, as it is mentioned later in the manuscript in lines 388-391.
- Lines 51-57 lack proper citation.
- Lines 96 and forward: A mention can be done to recent evidence of increased glomerular filtration.
- Lines 100-101: The vasodilatation does not cause hypovolemia, it causes relative hypovolemia. This is an important distinction because recently, some evidence indicates that a restrictive-fluid approach is not inferior when it is compared with the 30 cc/kg fluid resuscitation recommended by the guidelines (DOI:10.51684/FIRS.129271, DOI: 10.1056/NEJMoa2202707). Moreover, regarding the redaction of the paragraph: “Infusion of large volumes of crystalloids more promoting this phenomenon” it's not quite clear what phenomena it is referenced. It might be agreed that the infusion of large volumes of crystalloids is deleterious, but cite 17 does not explain this matter, neither the endothelial dysfunction nor the increase of Vd. Please add proper citation.
- Line 110-111: The administration of vasopressors during sepsis may contribute to the decrease of fluid administration which may lead to avoid the increment in Vd. Currently, there are no studies that report a decrease of tissue penetration of antibiotics in patients receiving vasopressors, additionally the reference 21 does not indicate that.
- Lines 117-122 lack proper citation.
- Lines 142-144 lack proper citation.
- Line 146. Antibiotic overexposure is also a problem in the dosing of betalactams. This has been observed in betalactams with high protein binding fraction (Ceftriaxone - Benzilpenicilin). Reference 29 corroborates this problem when the free fraction of this class of antibiotics has been measured. On this section can be mentioned the problem of increased glomerual filtration if not mentioned before (point 3). Evidence from DALI (ref 8) has shown that the renal failure confounds profoundly the PK/PD models and an important uncertainty in the levels and PK of the antibiotic occur, please mention.
9. Line 236: Information on microbiological methods for measuting antibiotics must be included. This easily implemented technique might be useful for not complexity labs or facilities with lower technology. See: doi: 10.1016/j.mimet.2016.07.020, DOI: 10.1080/1120009x.1989.11738901, doi: 10.1186/2050-6511-14-59.
10. Line 318-320: This is controversial and not evidence based, please add some references. The DALI study (ref 8) was not able to demostrate a difference between those two targets and in the multivariate models, both measures showed similar impact on the clinical outcome. Please mention it. Since this is the basis for the proposed diagram, please add more information.
- Line 264. In relation to the nephrotoxicity of beta lactams, this is a hard topic. The utilization of creatinine levels to define kidney damage could be misleading, and an oversimplification of this problem. I recommend taking a look at this reference https://doi.org/10.1007/s00134-022-06811-0. It may be important to indicate that the creatine levels in the critically ill does not relate with kidney dysfunction.
- The reference 114 has been already published (Intensive Care Med (2022) 48:1760–1771, https://doi.org/10.1007/s00134-022-06921-9), the investigators did not find a beneficial effect of model-informed precision dosing of beta-lactam antibiotics and ciprofloxacin in critically ill patients. Thus, it should be emphasized the need to identify other approaches to dose optimization, given the lack of strong evidence that supports TDM across the studies.
- I suggest to include the reference (https://doi.org/10.1186/s13054-022-04098-7), highlighting the fact that the use of tools that include TDM such as Autokinetics, failed to reach the primary outcome of pharmacokinetic target attainment in the first 24 hours after randomization for beta-lactam antibiotics. More importantly, the results of this study did not show clinically or statistically significant differences in terms of relevant clinical endpoints such as mortality, ICU length of stay and incidence of acute kidney failure between both arms (Autokinetics and control group). I recommend further explaining the discussion about the real utility of TDM and the target sub-groups of critically ill patients that may benefit more from the use of adaptive dosing adjustments using precision-based antimicrobial dosing. This editorial may be of help (https://doi.org/10.1007/s00134-022-06969-7).
- Regarding the proposed algorithm, there are some observations: 1) How does one define “bacteria with high MIC?. Further explain this phrase; 2) The abbreviature G- referring to gram negative has not been defined early in the text, it requires clarification; 3) In the severity assessment step, clarify if all three conditions must be fulfilled for TDM or just one of those mentioned.
Round 2
Reviewer 1 Report
This paper has been improved but remains very bad organised. It seems that authors want so say many things but the way they reported "consecutive concepts" is misleading. The reader jump from this to that and then back to this. Very confusing. Examples:
Page 3, line 102: describe APC here
Page 3, line 106-108: what do you mean with fluid resuscitation-induced vasodilatation? It doesn't make any sense.
Page 5, line 216-221: I don't really understand the reason that led you to report ECMO among considerations about PK alterations and kidney dysfunction.
Page 6, lines 282-283: That's not fully right. MIC may indicate susceptible and resistant strains leading to completely different therapeutic exposure target.
Page 6, lines 330-333: Again, it's unclear. I mean, If piperacillin increases creatinine secretion at the tubular level, than serum creatinine should decrease. Otherwise, if something competes for creatinine secretion, its blood value should rise.
Reviewer 2 Report
None additional.